

# Global Sensitivity Analysis and Adaptive Stochastic Sampling of a Subsurface-Flow Model Using Active Subspaces

Daniel Erdal[1] and Olaf A. Cirpka[1]

[1]University of Tübingen, Hölderlinstr. 12, 72074 Tübingen, Germany

**Correspondence:** Daniel Erdal (daniel.erdal@uni-tuebingen.de)

**Abstract.** Integrated hydrological modelling of domains with complex subsurface features requires many highly uncertain parameters. Performing a global uncertainty analysis using an ensemble of model runs can help bring clarity which of these parameters really influence system behavior, and for which high parameter uncertainty does not result in similarly high uncertainty of model predictions. However, already creating a sufficiently large ensemble of model simulation for the global
sensitivity analysis can be challenging, as many combinations of model parameters can lead to unrealistic model behavior. In this work we use the method of active subspaces to perform a global sensitivity analysis. While building-up the ensemble, we use the already existing ensemble members to construct low-order meta-models based on the first two active subspace dimensions. The meta-models are used to pre-determine whether a random parameter combination in the stochastic sampling is likely to result in unrealistic behavior, so that such a parameter combination is excluded without running the computationally
expensive full model. An important reason for choosing the active subspace method is that both the activity score of the global sensitivity analysis and the meta-models can easily be understood and visualized. We test the approach on a subsurface flow model including uncertain hydraulic parameter, uncertain boundary conditions, and uncertain geological structure. We show that sufficiently detailed active subspaces exist for most observations of interest. The pre-selection by the meta-model significantly reduces the number of full model runs that must be rejected due to unrealistic behavior. An essential but difficult part in
active subspace sampling using complex models is approximating the gradient of the simulated observation with respect to all parameters. We show that this can effectively and meaningful be done with second-order polynomials.

## 1 Introduction

Water flow in the subsurface is an integral part of the water cycle. In recent years, partial-differential-equation (*pde*) based integrated hydrological modelling, coupling flow in the subsurface and on the land surface, have become a rather standard
tool (Maxwell et al., 2015; Kollet et al., 2017). With the increasing computational power also the size of the models have increased (Kollet et al., 2010). However, increasing the size and/or complexity of a model usually also increases the number of (spatially variable) parameters of to the model. Identifying suitable parameter values from a limited number of observed data (i.e. calibration, inverse modelling, parameter estimation) is, and has been, a large topic in hydro(geo)logical modeling (Vrugt et al., 2008; Shuttleworth et al., 2012; Yeh, 2015). Related to the topic of model calibration, is the question how sensitive
certain parameters actually are to the observed data, and, hence, which parameters can and cannot be inferred from this data.



The latter is explored by sensitivity analysis (e.g., Saltelli et al., 2004, 2008). A clear separation in the sensitivity analysis literature is between local methods, in which parameters are varied about a fixed point, and global methods, which aim to explore sensitivities of the parameters across the full parameter space. The two approaches lead to identical results only when the dependence of the model outcome on the parameter values is linear. For hydrological purposes, the recent reviews of Mishra

et al. (2009), Song et al. (2015), and Pianosi et al. (2016) provide structured overviews and selection suggestions for the choice of an appropriate sensitivity-analysis method. There exists a large collection of different global sensitivity-analysis methods. Song et al. (2015) divides them into screening methods, regression methods, variance-based methods, meta-modeling methods, regionalized sensitivity analysis, and entropy-based methods, each of which contains multiple implementation variants. The popular method of Sobol (1993) indices is a typical example of a variance-based method.

A global sensitivity approach, that does not directly fit into any of the categories listed above but has recently gained increased attention, is the active subspace method (e.g., Constantine et al., 2014; Constantine and Diaz, 2017). The aim of the subspace method is to find the most influential directions in parameter space. An active subspace is defined by active variables, which are linear combinations of the investigated parameters. Along the active variables, the observation changes more, on average, than along any other direction in the parameter space. The method has mainly been applied to engineering-

related models (e.g., Constantine et al., 2015a, b; Hu et al., 2016; Glaws et al., 2017; Constantine and Doostan, 2017; Hu et al., 2017; Grey and Constantine, 2018; Li et al., 2019), however, recently it has also successfully been applied to coupled surface-subsurface flow simulations. Jefferson et al. (2015) used the coupled subsurface-land surface model ParFlow-CLM to study the sensitivity of energy fluxes on vegetation and land-surface parameters. Apart from deriving sensitivities, they showed that an active subspace for a model including subsurface flow existed. Jefferson et al. (2017) applied the active subspace

method to the same model (ParFlow-CLM) to study the sensitivity of transpiration and stomatal resistance on photosynthesis-related parameters. Actively considering the deeper subsurface (i.e. groundwater flow), Gilbert et al. (2016) used ParFlow in combination with active subspaces to study the effect of three-dimensional hydraulic-conductivity variations on cumulative runoff. They showed that the method of active subspaces can successfully be applied to complex subsurface-flow models. However, they also showed that an active subspace may not be well defined under unsaturated conditions and Hortonian flow.

A general problem when performing any type of global sensitivity analysis is the choice of how to sample the parameters. Apart from defining ranges and distributions of single parameters, which are unique to the problem at hand and can often be addressed by experts in the field, questions related to unfavorable parameter combinations are harder to deal with *a-priori*. Unfavorable parameter combinations may lead to a model behavior that is not observed in reality, such as severe floodings or strong droughts. In regionalized (a.k.a. generalized) sensitivity analysis, such parameter sets are classified as non-behavioral, and sta-

tistical differences between the behavior and non-behavioral parameter sets are sought (Spear and Hornberger, 1980; Beven and Binley, 1992; Saltelli et al., 2004). Another way of approaching the problem consists in discarding the non-behavioral parameter sets as unrealistic (or unphysical or model failures), hence performing some type of rejection sampling. The continuing analysis is then done on the remaining sets, i.e. using samples from a constrained joint parameter distribution. The clear drawback of this approach is that many, potentially expensive, model simulations may be performed and then discarded.



Recognizing that a sufficiently large set of model runs is needed for a reliable stochastic analysis, Song et al. (2015) discussed the use of meta-models in hydrological sciences. The underlying basic idea is to calibrate a computationally inexpensive model, denoted meta-, surrogate-, emulator-, or proxy-model, to the input and output data from a small set of complete model runs. The sensitivity analysis is then performed using the meta-model rather then the original model (Ratto et al., 2012). Razavi et al.

(2012) have reviewed different types of meta-models, among others, polynomials, multivariate adaptive regression, artificial neural networks, support vector regression, and, Gaussian processes. When using the active subspace method, a benefit is that a low-dimensional response surface (i.e. a meta-model) can be fitted between the derived active variables and the data (see Li et al., 2019). The better such a surface fits, the better the decomposition was. Apart from being easy to visualize (in case of 1 or 2 dimensions), the surface is also trivial to fit if, e.g., simple polynomials are considered. However, a problem of meta-modeling

is that any analysis done with the meta-model is only as good as the meta-model itself, and parameter sensitivities derived with the meta-model may be biased by the simplified input-output relationship.

In this paper, we use a meta-model, derived by the active subspace method, to pre-determine assumably behavioral parameter sets and perform the global sensitivity analysis with the full model using the pre-selected parameter sets. By this we aim at reducing the number of discarded simulations with the full model. We use two-dimensional active subspaces to derive

both multiple meta-models and sensitivity patterns for an integrated surface-subsurface flow model. The rest of the paper is structured as follows: Section 2 gives a general description of the methods applied in this study, Section 3 introduces the test case to which the methods are applied, Section 4 presents and discusses the result of both adaptive sampling and sensitivity analysis. The paper closes with general discussions and conclusions in Section 5.

## 2 Methods

### 2.1 Governing Equations and Simulation Code Used

Flow in the subsurface is computed using the software HydroGeoSphere (Aquanty Inc., 2015). Although HydroGeoSphere can simulate the entire terrestrial portion of the water cycle, we focus in this work on subsurface features. HydroGeoSphere provides a finite element solution of the 3-D Richards equation, here presented in a general form without explicit consideration of boundary conditions, with the purpose of facilitating the discussion of parameters later on.

$$S_s S_w(h)\frac{\partial h}{\partial t} + S_w(h)\theta_S\frac{\partial S_w(h)}{\partial t} = \nabla \cdot (\mathbf{K_s}k_r(h)\nabla(h+z)) + Q \tag{1}$$

in which $h$ [L] is the pressure head (i.e. the hydraulic head minus the geodetic height $z$ [L]), $S_w$ [-] is the water saturation, $\theta_S$ [-] is the effective porosity, $S_s$ [1/L] is the specific storativity, $\mathbf{K_s}$ [L/T] is the saturated hydraulic-conductivity tensor, $k_r$ [-] is the relative permeability, and $Q$ [1/T] represents sources and sinks. The retention and relative-permeability functions are computed using the standard Mualem van-Genuchten model (Mualem, 1976; Van Genuchten, 1980):





$$S_e = \begin{cases} [1 + (\alpha[h])^n]^{-m} & \text{if } h < 0 \\ 1 & \text{otherwise} \end{cases} \tag{2}$$

$$k_r = [1 - (1 - S_e^{1/m})^m]^2 S_e^{0.5} \tag{3}$$

in which $S_e$ [-] is the effective saturation, which relates to water saturation by $S_e = (S_w - S_r)/(1 - S_r)$, where $S_r$ [-] is the residual saturation. Further, $\alpha$ [1/L], $n$ [-], and $m = 1 - 1/n$ [-] are shape parameters. In terms of parameter values, this work

focuses mainly on the two shape parameters, the saturated hydraulic conductivity and specific storativity.

## 2.2 Derivation of the Active Subspace

In this section we consider a general function $f$ generating a scalar output and requiring an input parameter vector $\mathbf{x}$. In this paper, computing $f$ involves running HydroGeoSphere and extracting a wanted (scalar) output.

The basic idea of an active subspace decomposition is to find primary directions in the original parameter space, composed

of linear combinations of the parameters, along which the solution $f(\mathbf{x})$ changes on average more than in other directions. To avoid effects related to different dimensions and magnitudes of parameters, all parameters are shifted and scaled to the range $[-1, 1]$ prior to the following calculations:

$$\tilde{x}_i = 2 \frac{x_i - x_{i,min}}{x_{i,max} - x_{i,min}} - 1 \tag{4}$$

in which $x_{i,min}$ and $x_{i,max}$ are the lower and upper bounds of parameter $x_i$, and $\tilde{x}_i$ is the scaled parameter.

An active subspace for $f$ is then defined by the eigenvectors of the following matrix (Constantine et al., 2014):

$$\mathbf{C} = \int \nabla f(\tilde{\mathbf{x}}) \otimes \nabla f(\tilde{\mathbf{x}}) \rho(\tilde{\mathbf{x}}) d \tag{5}$$

with its eigen-decomposition:

$$\mathbf{C} = \mathbf{W \Lambda W}^{-1} \tag{6}$$

in which $\otimes$ denotes the matrix product, $\rho$ is the probability density function of the scaled parameters $\tilde{\mathbf{x}}$, the integration is per-

formed over the entire parameter space, $\mathbf{W}$ is the matrix of eigenvectors, and $\mathbf{\Lambda}$ is the matrix of the corresponding eigenvalues. Because $\mathbf{C}$ is symmetric and real, the eigenvectors contained in $\mathbf{W}$ are orthogonal to each other, $\mathbf{W}$ can be interpreted as a rotation matrix in parameter space, and the inverse $\mathbf{W}^{-1}$ is identical to the transpose $\mathbf{W}^T$. We perform the integration in Eq. (5) by the Monte Carlo method (Constantine et al., 2016; Constantine and Diaz, 2017):

$$\mathbf{C} \approx \frac{1}{M} \sum_{i=1}^{M} \nabla f(\tilde{\mathbf{x}}_i) \otimes \nabla f(\tilde{\mathbf{x}}_i) \tag{7}$$





in which $M$ is the number of samples used and $\tilde{\mathbf{x}}_i$ are independently drawn samples of $\tilde{\mathbf{x}}$. Now the aim is to find a subset of $n$ eigenvectors that sufficiently describe the relation between $\tilde{\mathbf{x}}$ and $f$ to create a decent low-order approximation $f(\tilde{\mathbf{x}}) \approx g(\mathbf{W}_n^T \tilde{\mathbf{x}})$. Here, $\mathbf{W}_n$ is the $m \times n$ matrix containing the eigenvectors with the $n$ largest eigenvalues. In our application, we choose $n = 2$.

For assessing the global sensitivity of each parameter in $\mathbf{x}$, we use the metric of Constantine and Diaz (2017), denoted the activity score $a_i$:

$$a_i = \sum_{j=1}^{n} \lambda_j w_{i,j}^2. \tag{8}$$

in which $i$ is the parameter index, $\lambda_j$ is the $j$-th eigenvalue and $w_{i,j}$ the value for parameter $i$ in the $j$-th eigenvector. Since the unit of the eigenvalues, and hence also of the activity score, is the square of the unit of the observation, we present in this work

the square root of the activity score rather than the activity score itself.

A major issue with computing an active subspace for a subsurface flow model, is that it requires the gradient of the target quantity $f$ with respect to all scaled parameters $\tilde{x}_i$ at all parameter values accessed, which is not readily available. A common workaround is to derive the gradients from a simple polynomial model between the model parameters and output. As our standard approach, similar to Grey and Constantine (2018), we fit a second-order polynomial to the data (gradient fit 1), but

we also test a second-order polynomial without cross-terms (gradient fit 2), and a linear model (gradient fit 3):

$$\text{gradient fit 1:} \quad \hat{f}(\tilde{\mathbf{x}}) = b_0 + \sum_{i=1}^{m} b_i \tilde{x}_i + \sum_{i=1}^{m} \sum_{j=i}^{m} b_{ij} \tilde{x}_i \tilde{x}_j \tag{9}$$

$$\text{gradient fit 2:} \quad \hat{f}(\tilde{\mathbf{x}}) = b_0 + \sum_{i=1}^{m} b_i \tilde{x}_i + \sum_{i=1}^{m} b_{ii} \tilde{x}_i^2 \tag{10}$$

$$\text{gradient fit 3:} \quad \hat{f}(\tilde{\mathbf{x}}) = b_0 + \sum_{i=1}^{m} b_i \tilde{x}_i \tag{11}$$

in which $m$ is the number of parameters. We determine the $b$-coefficients by standard multiple regression from an ensemble of

model runs. The gradient fit 1 requires $m^2/2 + 3m/2 + 1$ $b$-coefficients, the gradient fit 2 requires $2m + 1$ coefficients, and the gradient fit 3 only $m + 1$.

Our standard fit is the second-order polynomial with cross-terms. If the set of model runs is smaller than about twice the number of required $b$-coefficients, we use the gradient fit 2 excluding the second-order cross terms. The linear fit 3 implies that the gradient $\nabla f(\tilde{\mathbf{x}})$ is independent of the parameter values and the summation in Eq. (7) would become unnecessary. It can be

shown, that under these conditions the number of active subspace dimensions reduces to one, and the associated eigenvector is the gradient itself. A benefit of using higher-order polynomial expressions to obtain the gradients, is that multiple subspace dimensions can be calculated, which we utilize and show to be beneficial in the present work.



## 2.3 Definition of a Meta-Model Using Active Subspaces

With a functional active subspace decomposition, we may construct a low order approximation of the observation ($f(\mathbf{x}) \approx g(\mathbf{W}_n^T \tilde{\mathbf{x}})$). In this work we consider a third-order polynomial surface fitted to our two active variables. This surface is later used a a meta-model in the adaptive sampling scheme presented in the next section.

Towards this end, we construct the vector $\boldsymbol{\xi}$ of reduced parameters (active variables) from the matrix of eigenvectors $\mathbf{W}_n$ associated to the $n$ largest eigenvalues of $\mathbf{C}$:

$$\boldsymbol{\xi} = \mathbf{W}_n^T \tilde{\mathbf{x}} \tag{12}$$

and fit the full solution to a third-order polynomial:

$$g(\boldsymbol{\xi}) = \beta_0 + \sum_{i=1}^{n} \beta_i \xi_i + \sum_{i=1}^{n} \sum_{j=i}^{n} \beta_{ij} \xi_i \xi_j + \sum_{i=1}^{n} \sum_{j=i}^{n} \sum_{k=j}^{n} \beta_{ijk} \xi_i \xi_j \xi_k \tag{13}$$

which involves 10 $\boldsymbol{\beta}$ coefficients for $n = 2$. We judge the quality of the third-order polynomial meta-model by the Nash-Sutcliffe Efficiency:

$$NSE = 1 - \frac{\sum_{i=1}^{M} (g(\boldsymbol{\xi}_i) - f(\mathbf{x}_i))^2}{\sum_{i=1}^{M} (f(\mathbf{x}_i) - \overline{f(\mathbf{x})})^2} \tag{14}$$

where $M$ is the number of samples, $f(\mathbf{x})$ is the result of the HydroGeoSphere simulation and $\overline{f(\mathbf{x})} = M^{-1} \sum_{j=1}^{M} f(\mathbf{x}_j)$ is the ensemble mean. In principle, the NSE ranges from minus infinity to 1, with values close to 1 marking better behaving models.

An NSE-value smaller than zero would imply that taking the mean of the full model calculations performs better than the meta model; such behavior is excluded when performing a polynomial fit to the data. A variety of other quantification metrics can be found in the supplementary material.

## 2.4 Adaptive Sampling Using Active Subspaces

A key difficulty in running complex models with random parameters drawn from wide prior distributions, is that a significant
number of the resulting model simulations may show a behavior that is contradictory to the prior knowledge of the modelled system. Such non-behavioral runs should be discarded in subsequent analyses, which implies that running them was a waste of computational resources.

    An approach to limit the number of non-behavioral model runs could be adaptive sampling, in which a meta-model (i.e. a simplified, fast running, low-order approximation of the true model) is used first to predict whether a parameter-set is be-
havioral. In this study we utilize the ability of an active subspace to construct low-order meta-models between our unknown parameters, represented as active variables, and the chosen observations whose behavior we wish to control. In our application, we use the third-order polynomials of the active subspaces, explicated in Eq. (13), as meta-models and judge the goodness of the meta-model by the NSE in Eq. (14). Part of the reason for choosing to work with a polynomial meta-model based on the active subspace, rather than a more complex meta-model based directly on the parameters, is its ease of use. The derivations are




simple, the meta-model and its fitting are standard procedures, and, above all, visualization, and, hence, intuitive understanding of the result, is trivial. This makes it an attractive approach, also for practitioners and others less interested in meta-modelling theory.

The setup of an adaptive sampler using active subspaces consists of the following seven steps:

1. Run a first set of flow models with random parameters drawn from a wide distribution of plausible values. In this work we use 500 as our initial sample size and apply Latin Hypercube sampling.

2. For each of the behavioral targets, construct a sufficiently detailed active subspace. If the gradient computation allows it, several subspace dimensions would be better than one. Here we use two active subspace dimensions (i.e. two active variables), and, hence, our meta-model is a surface in the two-dimensional space of active variables. For each meta-model, an NSE-value larger than 0.7 is required to be considered for pre-assessing the behavior of new parameter sets in the following.

3. A new candidate of the full parameter vector is now drawn from the same initial distribution as used in step 1. This parameter vector is projected onto the active subspace(s), and the meta-model is used to approximate the behavior of all target predictions.

4. The new candidate will be accepted at stage one if any of the following criteria are met:

    (a) All approximated behavioral targets are on the permitted side of their limits.

    (b) While a target is on the non-behavioral side of the limit, it is within a reasonable distance of the limits. In practice, this is implemented as a linear decay function from 1 at the limit to 0 at an outer (user specified) point. If the decay function value is larger than a random number drawn from a uniform distribution, the candidate is accepted. This criterion is implemented to construct a soft region around the limits that accounts for the imperfection of the meta-model. The outlined approach is similar to the classical Metropolis-Hastings sampling (Metropolis et al., 1953; Hastings, 1970).

    (c) With a 10% probability, a candidate is accepted at stage one independent of its predicted performance. This criterion is included to make sure that we maintain a sufficiently good sample of the full parameter space, so that the re-calculation of the active subspace (see below) still sees the unwanted regions.

    If the candidate is stage-one rejected, repeat points 3-4 until a successful candidate is drawn.

5. For each stage-one accepted candidate, we run the full flow model to obtain the prediction of the real model (stage two).

6. After performing a predefined number (in our case 100) of flow simulations, re-calculate the active subspaces using all flow-simulation outputs obtained so far.





7. Repeat steps 3-6 until the sample size is large enough for the purpose of the stochastic modelling. This is a model-purpose specific choice, and can be done both on hard limits to the (stage-two) simulated data or on the number of flow model runs. Here, we require 10,000 runs of the flow model (i.e. 9,500 stage-one acceptances plus 500 initial samples).

While the active subspace(s) and the meta-model are deliberately constructed with an ensemble that includes non-behavior model runs, the stochastic analyses of the final ensemble is constrained to the behavioral runs. Also note that we use the meta-model only as a pre-selection tool.

## 3 Application to a Virtual Test Case

### 3.1 Description of the Domain

The test bed used in this paper is a steady-state flow model setup and run in HydroGeoSphere. It draws its main features from the catchment of the stream Käsbach in the Ammer-valley in southwestern Germany (Selle et al., 2013) with some simplified features. That is, the simulated domain is not meant as exact representation of the Käsbach catchment but contains enough details to be considered a realistic test for the proposed global sensitivity analysis method.

As illustrated in Figure 1, the subsurface model consist of 5 geological layers, representing the major lithostratigraphic units in the region. From the bottom to the top, these are: (1) the middle-Triassic Upper Muschelkalk formation, made of fractured-karstified limestone, (2) the lower upper-Triassic Lettenkeuper (Erfurt formation), made of clay-rich mudstones and carbonate-rock layers, (3) the unweathered middle upper-Triassic Gipskeuper (Grabfeld formation), made of mudstones and gypsum-bearing layers, (4) a weathering zone of the latter formation, and (5) Quaternary valley fills of unconsolidated sediments. A fault passes through the domain in the North-South direction, leading to offsets in the geological units. The geological base model resembles the regional model of D'Affonseca et al. (2018). Each layer is modeled as homogeneous unit.

The model domain measures about 4 km × 6 km at the widest places. It is discretized by 1,001,760 prism elements using 523,083 nodes and features a single main stream with four possible tributaries. The model is setup and run in transient mode with constant forcings until steady state is reached. Only the final time step (here after simulating $10^{10}$ seconds) is considered in the analysis.

The boundary conditions, setup to allow water to leave the domain both through the surface and the subsurface, are as follows: The bottom of the model features a Dirichlet-boundary with values read in from a larger-scale model of the region (D'Affonseca et al., 2018), but limited such that no hydraulic head at the bottom face can be higher than 5 meters below the model top. Streams in the model are modelled as drains, meaning that water can exit there when the hydraulic head at the assigned stream nodes exceeds a value 1 cm above the surface elevation. This implies that all streams are either inactive or gaining, whereas losing conditions are excluded. A similar drain boundary, but with much higher exit head (fixed at 0.2 meters above land surface for all simulations) is also considered on all non-stream nodes in the upper most layer, to allow water to leave the domain in case of flooding. The last outflow boundary in the model is a Cauchy-boundary at the southern vertical wall of the model. To avoid long run times and complications of complex top soils, which are unimportant once steady





**Figure 1.** Illustration of the modelled catchment, including important features and surrounded by an explicit view on the different geological features.

state is reached, the top of the HydroGeoSphere model is 1 meter below land surface. Flow across the top boundary is only incoming and modelled as a Neumann-boundary corresponding the to steady-state groundwater recharge. The recharge varies with land-use, split into three categories: cropland, grassland, and forest, in which urban areas are treated as grassland.

## 3.2 Virtual Observations

5 For the evaluation of the model sensitivities, we consider four observation quantities: (1) the discharge almost at the outlet of the catchment (gauge C in Figure 1), (2) the net sum of the fluxes across the bottom and side subsurface boundaries, (3) the groundwater table of the uppermost aquifer, measured in 199 observation wells throughout the catchment, and (4) the



groundwater residence time in the major geological layers. The latter is representative for transport and differs in its sensitivity from hydraulic heads (e.g., Cirpka and Kitanidis, 2000). The time that a solute parcel stays in a particular geological formations may be indicative for reactive transport (e.g., Sanz-Prat et al., 2016; Loschko et al., 2016; Kolbe et al., 2019).

We computed the geological-unit specific residence time using the visualization software TecPlot360, considering 199 par-
ticles that each start in an observation well, 5 meters below the groundwater table. We separate the discrete particle tracks into the segment spent in each geological layer and compute the total residence time per layer as the mean over all 199 particles. It should be noted that this way of computing travel times is slightly imprecise, as the velocity-field output of HydroGeoSphere is non-conforming and TecPlot particle tracking is not primarily designed for quantitative outputs. However, the purpose of the residence-time calculation is not an exact prediction of exposure times in the real Käsbach catchment but rather a qual-
itative transport indicator. As we use the same method with the same numerical parameters on the same grid in all model runs of our stochastic ensemble, we believe that the associated variability of computed residence times is good enough for the inter-comparison between different stochastic runs.

### 3.3   Stochastic Treatment of the Geological and Hydraulic Parameters

In the setup used in this paper, 32 parameters related directly to the flow model are randomized. All parameters are sampled
from a uniform distribution. Table 1 list the corresponding parameter bounds. For each of the geological layers, the uncertain parameters are: the horizontal saturated hydraulic conductivity, the anisotropy ratio of horizontal to vertical conductivity (except for the Muschelkalk limestone and the Quaternary fillings), the two van-Genuchten parameters $\alpha$ and $n$, and the specific storativity. Further, the bottom Dirichlet-boundary, the reference head at the Cauchy boundary, and the head at the stream drain boundaries are drawn from uniform distributions.

Besides these material properties of the lithostratigraphic units and boundary conditions, also the exact subsurface structure is uncertain. To address this uncertainty, we drew three parameters controlling the size of the main geological layers from uniform distributions: the vertical offset of the fault running north-south through the domain (see Figure 1), the thickness of the Lettenkeuper (expressed as a difference to the base value in Table 1), and the thickness of the weathering zone in the Gipskeuper, in which the latter has the thickness of the Gipskeuper itself as upper bound. An example of the variability in
the subsurface can be seen in Figure 2, where six realizations of the Lettenkeuper layer are shown. Finally, also the recharge fluxes are randomized. For each of the three different land-use types discussed above, we draw random values of groundwater recharge in each sample, based on a large collection of 1-D simulations of the missing first meter of the subsurface, using different soil structures, plants parameters, and top boundaries. The resulting ranges are shown in Table 1.

The stochastic engine for HydroGeoSphere is setup in and controlled by Matlab. The full stochastic suite is run on a mid-
range cluster with 20 nodes, each featuring 2 Intel Xeon L5530, 8 core, 2.4GHz processors with 72Gb RAM. The 10,000 samples discussed later took on this setup about 96,000 CPU-hrs (~11 CPU-years), corresponding to 300 wall-clock hours (~12 days).





**Table 1.** Sampling ranges for of stochastic parameters considered.

| Parameter → | Conductivity | | Anisotrophy ratio | | $\alpha$ | | $n$ | | $S_s$ | |
|---|---|---|---|---|---|---|---|---|---|---|
| | (m/s) | | (-) | | (1/m) | | (-) | | (1/m) | |
| Layer | Min | Max | Min | Max | Min | Max | Min | Max | Min | Max |
| mo | $10^{-7}$ | $10^{-5}$ | 1 | 1 | 0.5 | 5 | 1.5 | 9 | $10^{-6}$ | $10^{-4}$ |
| ku | $10^{-8}$ | $10^{-5}$ | 1 | 50 | 0.5 | 5 | 1.5 | 9 | $10^{-6}$ | $10^{-4}$ |
| km1 | $10^{-9}$ | $10^{-7}$ | 1 | 50 | 0.5 | 5 | 1.5 | 9 | $10^{-6}$ | $10^{-4}$ |
| km1-w | $10^{-7}$ | $5 \times 10^{-5}$ | 1 | 50 | 0.5 | 5 | 1.5 | 9 | $10^{-6}$ | $10^{-4}$ |
| Q | $10^{-7}$ | $10^{-5}$ | 1 | 1 | 0.5 | 5 | 1.5 | 9 | $10^{-6}$ | $10^{-4}$ |

| Parameter | Min | Max | Parameter | Min | Max |
|---|---|---|---|---|---|
| Bottom Head Offset (m) | -5 | 5 | River Bed Thickness (m) | 0.005 | 0.2 |
| Fault Offset (m) | 0 | 100 | Recharge Grass (mm/year) | 80 | 13ß |
| Contact Offset ku-km1 (m) | -20 | 20 | Recharge Crop (mm/year) | 100 | 150 |
| km1-w Thickness (m) | 5 | 50 | Recharge Forest (mm/year) | 100 | 150 |
| Cauchy Boundary Head (m) | 335 | 355 | | | |

mo: Upper Muschelkalk, ku: Lettenkeuper, km1: Gipskeuper, km1-w: weathered Gipskeuper, Q: Quarternary fillings.

## 3.4 Definition of Behavioral Targets

In the present work, we define five behavioral targets that are all based on expert knowledge about the modelled catchment. In the following, we specify the behavioral target values and the point of maximum deviation from that target for which we may probabilistically accept a simulation run, denoted "outer point":

**Limited flooding** Flooding, here viewed as water leaving the domain through the top drain at the surface at places outside of the streams, occurs in the model at a hydraulic head of 0.2 m above the land surface. The total flodding in the domain should not exceed $2 \times 10^{-3}$ m³/s (outer point $4 \times 10^{-3}$ m³/s). Some flooding is seen as acceptable, as it may occur in low-land areas next to the streams, which we don't model in great detail.

**Minimum flow in the main stream** At measurement gauge C (Figure 1), the stream should be fully developed, which we
define as a discharge larger than $5 \times 10^{-3}$ m³/s (outer point $3 \times 10^{-3}$ m³/s). This reference value is picked based on experience with the model domain and the known range of annual mean recharge.

**Minimum flow in stream B** Knowing that stream B produces flow, a minimum flow is set to $1.0 \times 10^{-6}$ m³/s (outer point $5.0 \times 10^{-7}$ m³/s ).

**Maximum flow in stream A** The stream residing on the steep eastern side of the hill is known to only produce flow under
15 extreme conditions. Hence, at steady state the flow in stream A should be minimal. The maximum accepted flow is



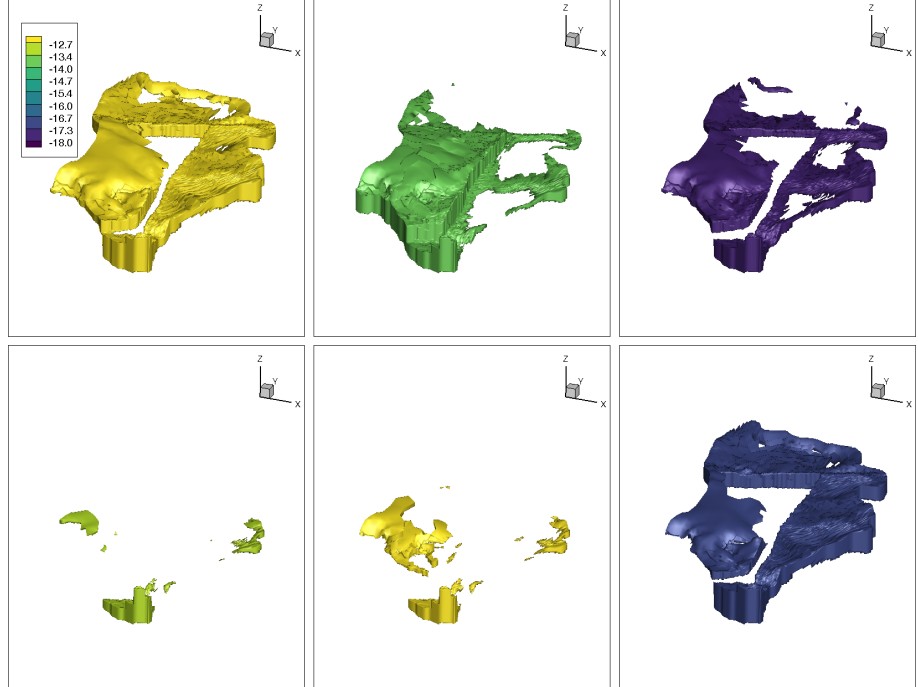

**Figure 2.** Six examples of distinctively different realizations of the geological layer Lettenkeuper. Color shows the natural logarithm of the saturated hydraulic conductivity.

therefore set to $1 \times 10^{-3}$ m³/s (outer point $2 \times 10^{-3}$ m³/s). A small flow is considered acceptable, since it may occur in the regions close to the main stream and hence be a reflection of model discretization rather than subsurface setup.

**Ratio of total stream discharge to total incoming recharge**  It is known that the catchment in question loses a notable amount of its water to the subsurface. A rough estimate is that, in the real catchment, the stream flow amounts to ≈40% of the incoming water. Based on this, we require that an acceptable model should have between 25% and 60% (outer points 20% and 75%) of its net recharge reaching the streams.

In a preliminary test of a model very similar to the one used here, and with the same randomized parameters, we performed Monte Carlo simulations of the full model without preselecting presumably behavioral parameter sets. Here about 75% of a total of 10,000 runs had to be discarded only due to severe flooding. This highlights that it is highly beneficial if the sampling is targeted only to simulations that show a response that is, within reason, representative for the modelled domain. In the case of flooding in the domain, it is not a single parameter that controls this behavior, but a complex relation between many parameters, deeming *a-priori* decisions about behavioral parameter ranges unfeasible.





## 4  Results

To allow the reader to better see the 3-D structures in the results presented here, the main results can be viewed in a plug-and-play app designed for Matlab, denoted "Active Subspace Pilot", which is available as supplementary information to this publication.

### 4.1  Adaptive Sampling

The effect of using the active subspaces as a sampling strategy for the flow simulations can be seen in Figure 3, showing the marginal distributions of the nine parameters that were most influenced by the sampling strategy. The blue bars are histograms of all parameter sets selected for full model runs, whereas the red bars are histograms of the behavioral parameter sets.

The blue bars of Figure 3 clearly show that already the pre-selection using the meta-model avoids certain regions of the parameter space. In particular, the two parameters related to the weathering zone of the Gipskeuper (conductivity and thickness of km1-w in Figure 3) show a preferential sampling for a thick and highly conductive layer. Similar preferences are seen in the Lettenkeuper and the Gipskeuper (ku and km1). This is contrasted by the deeper subsurface, where the Muschelkalk (mo) shows a preference towards low conductivity and the offset of the fault is preferably sampled at smaller values (which decreases the size and connectivity of the Muschelkalk layer). By selecting high conductivity values in the Quaternary and weathered Gipskeuper layers, chances of floodings are reduced. Further, the high conductive near-surface and middle-deep layers serve to transport water towards the streams. The smaller and less conductive deep subsurface in combination with higher bottom pressures, on the other hand, serves to inhibit exiting water through the bottom. Hence, the posterior sample shows exactly the behaviors required by the targets. This suggests that the sampling strategy has been successful.

The red bars in Figure 3 show the marginal posterior distribution of parameters used for the sensitivity analysis. This corresponds to a selection of simulations that are strictly better than the mean of the targets and their outer point (see above). Hence, this selection is deterministic with a hard limit. It is obvious that the pre-selection and final selection of parameters are similar, but the stricter sample used in the sensitivity analysis has fewer members. The blue bars in Figure 3 comprises 10,000 samples, while the red bars comprises a subset of 4,533 samples. Part of the reason for this rather larger difference, is that the active subspace sampler is only an approximation. More so, we have deliberately relaxed the criterion for accepting a parameter set by including a range around the target, and, for the full run to include a 10% acceptance independent of the meta-model prediction. In a comparable setup, we sampled 10,000 parameter-sets without pre-selection, and out of those, only 588 were acceptable with the strict criterion used here. Hence, the improvement when using the active subspace sampler is clearly notable.

Figure 4 shows the performance and development of the active subspaces for three representative targets. Here, the x- and y-positions of the markers are the values of the two active variables, respectively, while the color indicates the magnitude of the corresponding observation. The first row of sub-plots shows the initial sample of 500 model runs, the second row the first 1,000 runs, and the third row the final ensemble of 10,000 runs. In all scatter-plots the target observation varies significantly along the first active variable (x-axis), but that there is also notable dependence on the second active variable (y-axis). The





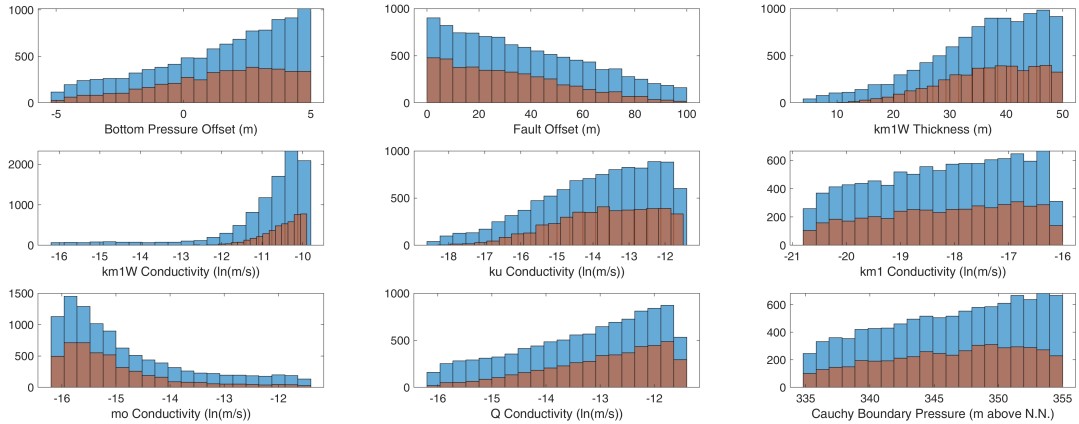

**Figure 3.** Marginal posterior distributions of the parameters influenced by the adaptive sampling. Blue bars show the sampled posterior and red bars the constrained posterior sample used in the sensitivity analysis.

latter suggests that it is appropriate to consider more than one active variable in the sampling procedure. Further, as indicated in the title of the subplots, the Nash-Sutcliffe Efficiency (NSE) for fitting the meta-models (Section 2.3) between the active variables and the data, are high, indicating that the active subspace decomposition has worked well. This is also exemplified in Figure 5, which shows the flooding observation and the fitted meta-model together with the corresponding error.

Figure 4 also shows notable differences between the active subspace constructed on the initial 500 runs (first row) and that after adding 500 actively sampled runs (second row). For example, for the ratio target (middle column in Figure 4), the orientation of the subspace changes, which is indicative of changes of weights within the active variable. We also see that the third-order meta-model used for the active subspace sampler fits the data better after extending the ensemble to 1000 members, as indicated by the NSE-values. By contrast, extending 1,000 to 10,000 pre-selected runs (second and third row in Figure 4),

does neither change the subspaces nor the surfaces of the meta-models in a significant manner, implying that the ensemble with 1,000 runs (500 initial plus 500 based on active-subspace sampling) already does a good job.

### 4.2    Global Sensitivity Analysis Using Active Subspaces

In this section, we use the active subspace method to analyze parameter sensitivities. In this analysis we only consider the behavioral parameter sets. That is, the red bars in Figure 3 define the probability density $\rho$ considered in Eq. (5).

The aim of a sensitivity analysis is to identify how influential individual parameters of a model are on (a set of) observations. In global sensitivity analysis, this is evaluated over the entire parameter space. We start the discussion with the least influential parameters in our application. None of the observations considered depend on the two van-Genuchten parameters $\alpha$ and $n$ or the specific storativity of any lithostratigraphic unit to a significant extent. The specific storativity is known not to affect steady-state flow at all. Also the van-Genuchten parameters are most important in transient flow in the unsaturated zone, or




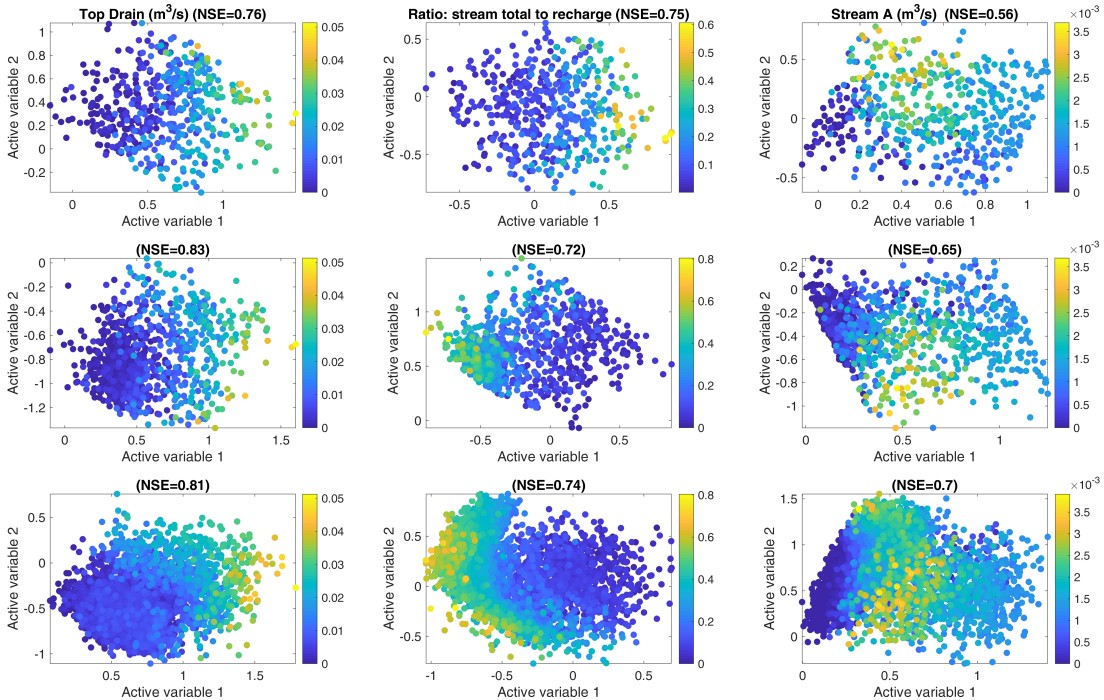

**Figure 4.** Two-dimensional active subspaces for 3 of the behavioral targets. The observed values are illustrated by the color. The first row shows the initial 500 samples, the second row the first 1,000 samples and the third row the full 10,000 samples. NSE values in the titles correspond to fitting the third-order meta model according to Eq. (13) to the data.

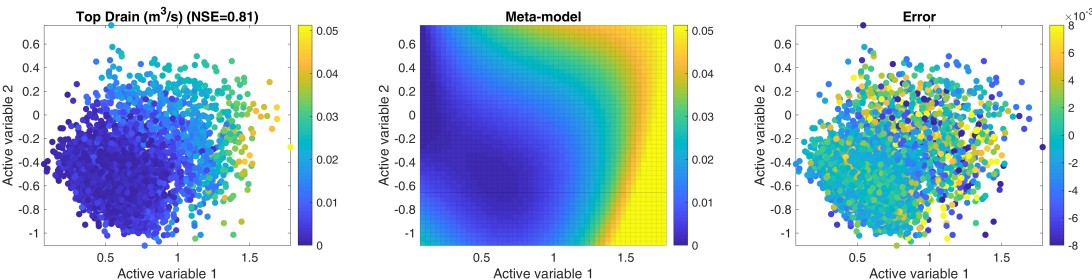

**Figure 5.** An example of the performance of the adaptive sampler for the flooding observation. The first plot show the data, the second plot the third-order polynomial meta-model fitted to the 10,000 observations in the first plot. The third plot shows the error between the true observations and the meta-model fit.

when there is a significant lateral flow component therein. Neither is the case in our application. That is, these result were to





be expected. Similarly, the sensitivity to the horizontal-to-vertical anisotropy ratio in all formations was small throughout the tests.

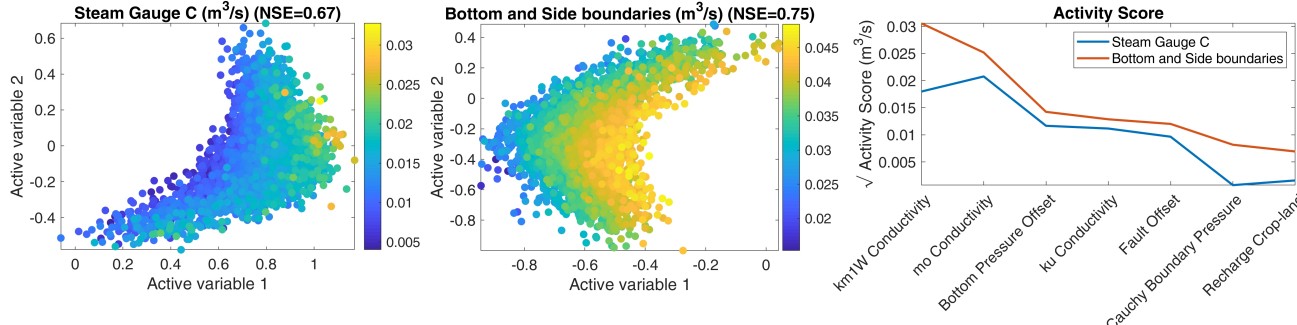

**Figure 6.** Active subspaces and square root of activity scores for the observations "Stream discharge at Gauge C" and "Net flow across subsurface boundaries". Please note that the plot is limited to the 7 most important parameters.

We now consider the significant parameters. Figure 6 shows the dependence of the targets on the two subspaces considered and the activity scores of all parameters for the discharge observation at Gauge C and the net flow across the subsurface

boundaries. As can be seen from the NSE-value in the titles, the active subspace decomposition works well for two active variables. We can also see that the sensitivity patterns for the two observations are rather similar: The hydraulic-conductivity values of the upper most and lowest geological units (km1-w and mo, respectively) are the most influential parameters. This is likely so because they decide the partitioning of the water between the surface streams and the subsurface boundaries. Interesting to note is that the applied strength of recharge does not show a high importance for the discharge in the streams,

implying that the partitioning of the water is more important than the actual net input when it comes to determining steady-state flow in this model.

Figure 7 shows the activity scores for the hydraulic head in the upper most aquifer in the 199 observation wells. Here we see that the sensitivity patterns differ among the different wells. However, we can classify the observation wells into three clusters: 1) those with a high sensitivity for the fault offset and the hydraulic conductivities in the Lettenkeuper (ku) and Muschelkalk

(mo) (marked in black); 2) those with high sensitivities to parameters related to the Gipskeuper and its weathering zone (marked in blue); and 3) those for which no particular sensitivities were found (marked in red). While the separation is not perfect, Figure 7 shows that there are only few overlaps. When plotting the spatial location of the different wells and their respective category (right plot in Figure 7), a clear pattern emerges. Almost all blue wells with high sensitivity to Gipskeuper-related parameters are placed in regions with Gipskeuper being present (north and east in the catchment), and, similarly, the black wells sensitive

to Lettenkeuper are located where we would expect the groundwater table to be found in this geological formation (western part of the catchment). The non-sensitive red wells are all placed close to the stream where hydraulic head is controlled by the stream stage rather than the properties of the geological layers. All in all, we can state that the method of active subspaces generates plausible results for groundwater observations in a complex geological setting.





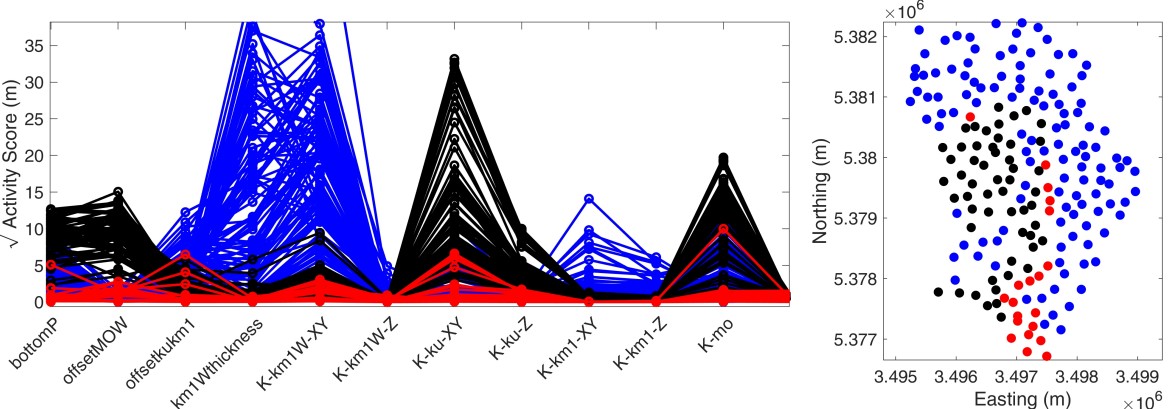

**Figure 7.** Square root of activity score for 199 wells (left) and their placement in the catchment (right). Please note that the activity score plot is limited to the 12 most important parameters.

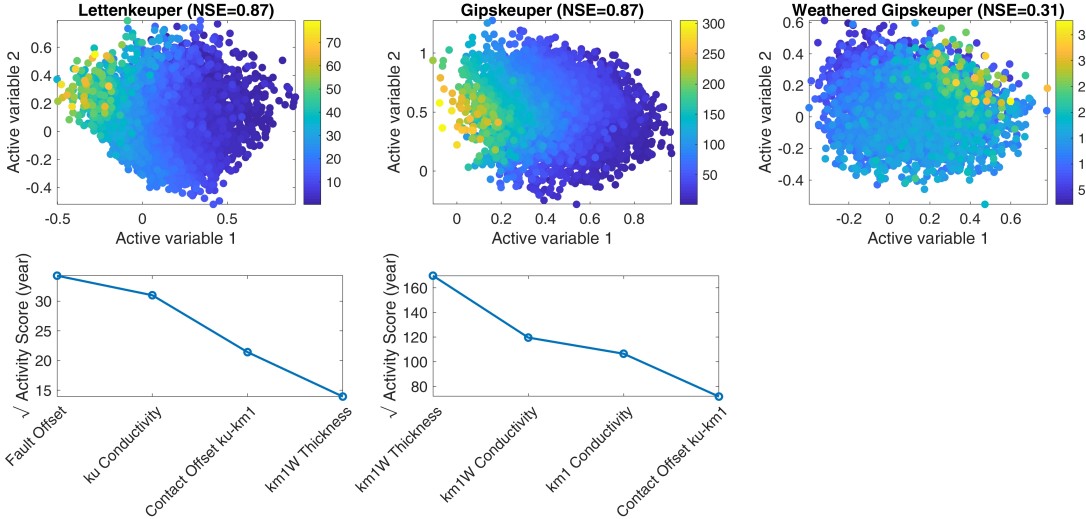

**Figure 8.** Two-dimensional active subspaces (top row) and square root of activity scores (bottom row) for residence time (years) in three geological units. Please note that the activity score plots are limited to the 4 most important parameters, and that due to the poor active subspace decomposition, this plot is not shown for the last column.

As a last observation, we consider the total residence time in the geological layers of the aquifer, which may be a relevant proxy for applications to reactive transport. Figure 8 shows the associated dependence of the targets on the two subspaces considered and the activity scores of all parameters. For the total residence time in the Lettenkeuper (ku) and Gipskeuper (km1), the corresponding meta-models using two active subspaces performed fairly well, as seen in the NSE-metric. This is

5  not the case for the total residence time in the weathering zone, where the cubic meta-model with two active variables achieved





a very low NSE of 0.3 (Figure 8, third column). Because of the bad fit of this observation, we don't show the associated activity-score plot. For the travel time through the other two layers, it is not so surprising that the hydraulic conductivity of the actual layer together with the parameters controlling the thickness of the layers, i.e. the fault offset for the Lettenkeuper and the thickness of the weathering zone for the Gipskeuper, are the controlling parameters. More interesting is that the hydraulic

conductivity of the weathering zone also plays a major role for the travel time through the non-weathered Gipskeuper (Figure 8, second row, second column). Hence, a good prediction of how long a water parcel stays within the unweathered Gipskeuper requires a good understanding of its weathered layer. The latter may be understood by the partitioning of water through the weathered and unweathered parts of the Gipskeuper. Increasing the hydraulic conductivity of the weathering zone leads to smaller volumetric fluxes through the unweathered Gipskeuper and thus lower velocities and larger residence times within this

unit.

## 4.3  Gradient Approximation in the Derivation of the Active Subspaces

In difference to previous works with active subspaces in subsurface flow, we approximate the gradient $\nabla f(\tilde{\mathbf{x}})$ from a second-order polynomial fit of $f(\tilde{\mathbf{x}})$ rather than a linear one. To test the effect of this approach, we compare the activity scores as well as Nash-Sutcliffe Efficiency (NSE) of the associated meta-models using the three different polynomial fits to obtain the

gradients discussed in section 2.2: (1) the full second-order model, (2) second-order approximation without cross-terms, and (3) a linear approximation.

To test the consistency of the results, we drew 2,500 samples in 1,000 repetitions using classic bootstrap-sampling without replacement from the original sample (4,533 members). In each repetition, we computed the activity scores and NSE of the meta-models based on the three gradient approximations. Figure 9 (left) shows the activity scores of the parameters applying

the gradient approximations for the discharge at Gauge C. Clear differences in the rankings among the different gradient approaches are obvious. The differences are the strongest between the linear approximation (which also reduces the number of possible active subspaces to one) and the two second-order approximations. Including the cross-terms in the second-order approximation increases all relevant scores with the biggest relative effect on the relevance of the hydraulic conductivity of the Lettenkeuper (ku).

To evaluate the goodness of each approximation, Figure 9 (right) shows the NSE resulting from fitting a third-order polynomial between the active variable(s) and the original data. It is obvious that the higher-order polynomial gradient-approximations are doing much better. It is important to keep in mind that we use the gradient fit only to approximate the gradients in Eq. (5), while the actual meta-model requires a second polynomial fit. The higher NSE-values of the more complex gradient fits thus indicate that they lead to active subspaces that are more indicative. As the linear gradient fit allows only the computation of a

single active subspace, the meta-model is indeed simpler. Including the second-order cross-terms seems to enrich the variability of the gradients over the parameter space, causing a separation of active subspaces that cover a wider range of parameter values. The results shown in Figure 9 give confidence that more complex gradient models are better if the data set is large enough to constrain all coefficients of such a model. Very similar results are found for all other behavioral targets, and the corresponding figures can be found in Appendix A.





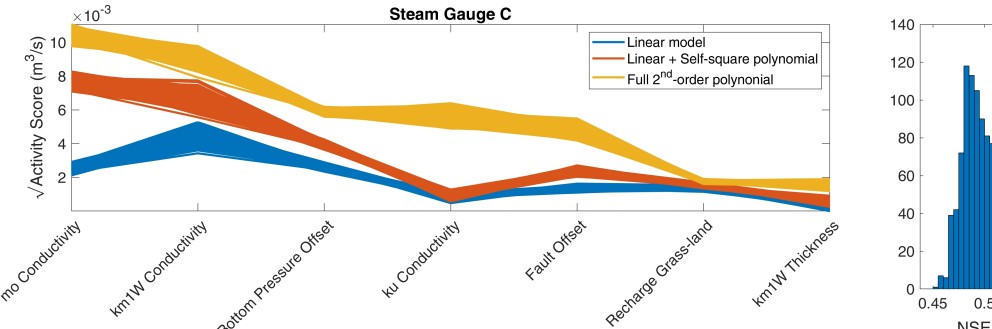

**Figure 9.** Square root of activity score and corresponding NSE for flow at gauge C using different approximations to compute the gradients. Each approximation is used to compute 1,000 active subspaces based on a bootstrap-resampling with 2,500 samples from the original 4,533. Only the 7 most important parameter are shown in the activity score plot.

## 5 Discussion and Conclusions

In this work we have applied the method of active subspace to an integrated hydrological model of a small catchment with focus on subsurface flow. We used active subspaces to construct meta-models with two active subspaces rather than 32 uncertain parameters. The meta-model was used to constrain the stochastic sampling of the parameter space to five behavioral conditions.

The active subspaces of the accepted full model runs were used to compute the global sensitivity of four modeled observations to the parameters. The sensitivity analysis showed that hydraulic-conductivity values of the major layers are important, but also their physical extent. However, depending on the location and type of observations, different sensitivities were found. This highlights the well-known fact that multiple, dissimilar observations are needed to constrain uncertain variables of a catchment model. In the adaptive sampling we learned that certain combinations of unfavorable parameter values where clearly avoided.

Most of the non-behavioral parameter combinations were not obvious beforehand, but could be identified by applying the meta-model, which significantly improved the sampling efficiency.

The choice of meta-model used in this work (third-order polynomial of the first two active subspace dimensions) was somewhat arbitrary. The number of different meta-models applied to hydrological problems is large (Razavi et al., 2012). Our guiding principles in selecting the meta-model were the good fit to the data, the ease in application, and the comprehensibility

for more practice-oriented users. While many state-of-the-art meta-models can be rather complicated, a surface depending on two dimensions is easy to understand, trivial to visualize and, hence, also allows a qualitative judgments by the user. We also performed preliminary tests using support-vector machines (results not shown), leading to results very similar to those of the active subspaces, but the method is more complicated to comprehend.

Choosing a low-order polynomial as meta-model implies smoothness and, hence, the meta-model does not exactly fit all

model runs. Razavi et al. (2012) argued that meta-models for computer simulations should always be exact because the computer simulations themselves are deterministic. We prefer the inexact model nonetheless because our meta-model is based on a limited number of active subspace dimensions. We explicitly ignore the other dimensions. If the considered subspace is





well determined, the ignored dimensions will still be present, but their effect can then be interpreted as noise. Hence, a smooth meta-model depending on a reduced-dimensional parameter set is applicable even though the computer simulations themselves are deterministic.

Even though the efficiency of the active subspace sampler is higher than a rejection sampler without pre-selection, the
rejection rate of $\approx 55\%$ is still rather high. This could of course be strongly decreased by setting the allowed soft targets to become harder. Such an approach would be appropriate if the main aim is to obtain as many behavioral parameter sets with the least effort, but we deliberately wanted to explore the behavioral boundaries of the parameter space, which requires stepping across that boundary. The choice of tuning parameters made in this work was made ad hoc and based on experience with the model domain and a qualitative assessment of the resulting surfaces. Better heuristic statistics could be implemented, which
could possibly further increase the efficiency of the sampler.

Overall, we draw the following conclusions from this work:

1. The method of active subspace can be applied with little effort and good results to complex subsurface flow and transport problems. This holds not only when subsurface properties are uncertain, but also the geometries of geological units and boundary conditions.

2. The two-stage rejection sampling using a meta-model based on the active subspaces can drastically decrease the number of simulations needed to obtain a certain number of behavioral simulations. An additional positive aspect for the application by practitioners is the ease of visualization and intuitive understanding when using a one- or two-dimensional active subspace.

3. Using a quadratic rather than linear fit to estimate the gradients in the construction of the active subspace resulted in a
much improved subspace decomposition. It is also a prerequisite to construct more than one subspace dimension.

*Code and data availability.* The supplement includes the Matlab R2019a code "Active Subspace Pilot" to visualize the active subspaces and scores for all model runs. This code is written as Matlab App and includes also the data.

## Appendix A: Performance of Different Gradient Approximations

This appendix contains additional plots showing the performance of the different gradient approximations for the four be-
havioral targets not presented in the main article. Each approximation is used to compute 1,000 active subspaces based on a bootstrap-resampling with 2,500 samples from the original 4,533 ensemble members. This is true for all observations apart from for the flow across the top boundary, where the full ensemble is used to draw samples from. Only the 7 most important parameters are shown in each activity-score plot.





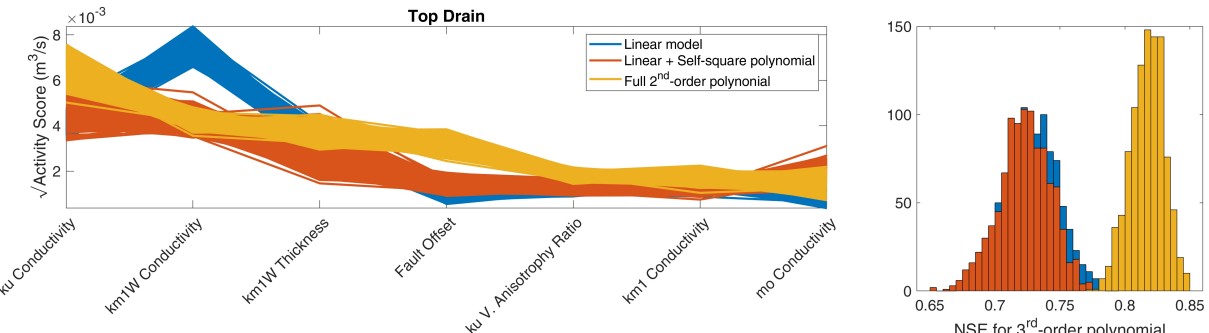

**Figure A1.** Square root of activity score and corresponding NSE for flooding flow across the top boundary using different approximations to compute the gradients. In this case, the bootstrap sample is drawn from the full sample population of 10,000.

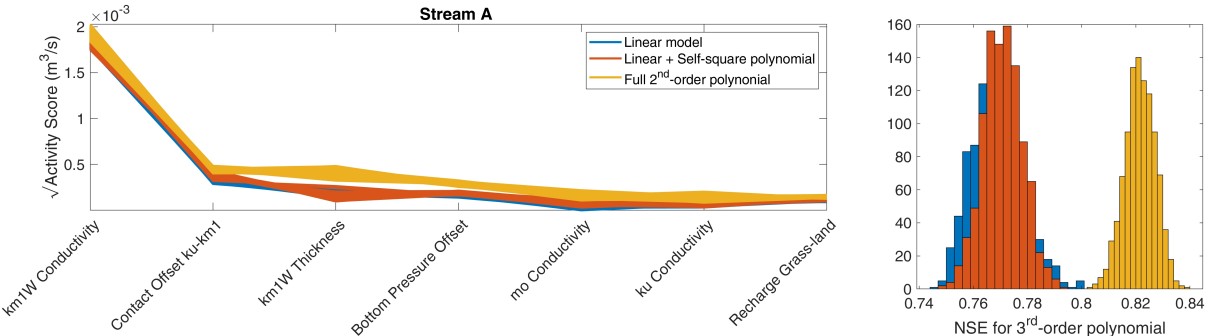

**Figure A2.** Square root of activity score and corresponding NSE for flow in Stream A (see Figure 1) using different approximations to compute the gradients.

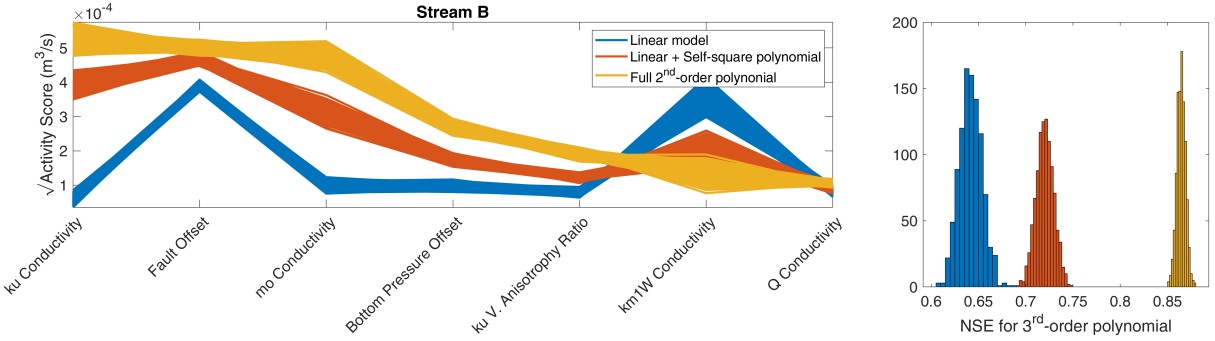

**Figure A3.** Square root of activity score and corresponding NSE for unwanted flow in Stream B (see Figure 1) using different approximations to compute the gradients.





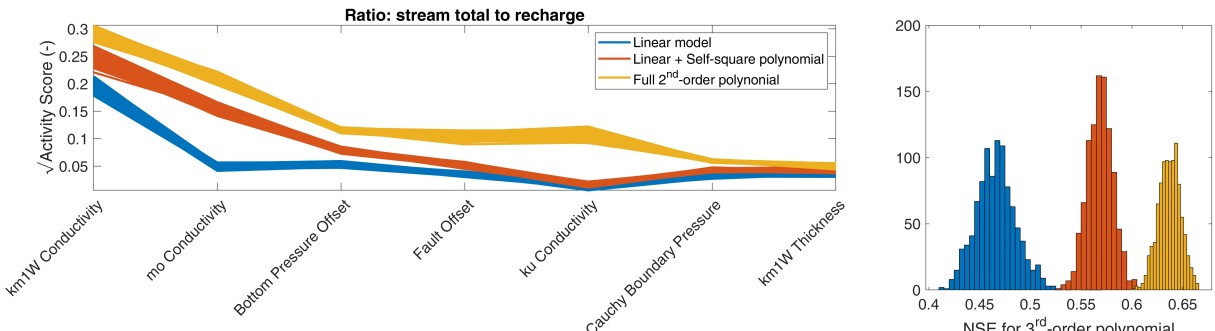

**Figure A4.** Square root of activity score and corresponding NSE for the ratio of stream discharge to incoming recharge using different approximations to compute the gradients.

*Competing interests.* No competing interests

*Acknowledgements.* This work was supported by the Collaborative Research Center 1253 CAMPOS (Project 7: Stochastic Modeling Framework of Catchment-Scale Reactive Transport), funded by the German Research Foundation (DFG, Grant Agreement SFB 1253/1).





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
