# Peer review of "Global Sensitivity Analysis and Adaptive Stochastic Sampling of a Subsurface-Flow Model Using Active Subspaces"

_Hydrology and Earth System Sciences, 2019_

## Referee Comment (RC1) · Anonymous Referee #1 · 3 May 2019

In this paper, the authors used active subspace for global sensitivity analysis and stochastic sampling of a subsurface-flow model. The topic is interesting and suitable for this journal. The adopted method also seems to be efficient and effective. However, some improvements are still required for final publication.

1. There are many useful methods for global sensitivity analysis as stated by the authors in the introduction. Why the authors use the activity score (equation (8))? Can this metric provide more reasonable results for the studied problem than other metrics? Please make an explanation.

2. The authors use active subspace to construct a meta-model in order to select the be-

havioral samples with a lower computational cost. In this work, only 2 active variables are used to construct the meta-model. Why do the authors use 2 active variables? As my understanding, more active variables will make a better meta-model. Do the authors think that 2 active variables are enough to construct an accurate meta-model? Please make an explanation.

3. The authors use meta-model to select behavioral samples. Since the meta-model only is an approximation of the real model, an error will arise when using meta-model. To construct the meta-model within active subspace, the authors first need to construct an active subspace. This requires the gradients of the model output with respect to the input parameters, which is a computational-demanding task. The authors fit a polynomial to approximate the real model, then obtain the gradient based on the polynomial. This approximation will cause another error. That is to say, there are two kinds of errors caused by approximation (meta-model) in the whole procedure. Since these two kinds of errors are not independent, the final error may be amplified. It will be better to analyze how these two errors will affect the final error of results.

4. For the meta-model within active subspace, the authors use third-order polynomial. Why do the authors use third-order? Is third-order enough for the meta-model? Some explanations should be given to convince readers.

5. In page 7(step 2), why do the authors construct an active subspace for each behavioral target? For a scaler model output, there is only one active subspace. Different behavioral targets can be defined based on the same model output. And in this situation, it will be not necessary to construct an active subspace for each behavioral target. I think constructing an active subspace for each model output will be reasonable.

6. In page 7(step 5), the accepted sample based on the meta-model is used for running the real model and obtain the prediction. Since the meta-model only is an approximation of the real model, the accepted samples based on the meta-model may not be really accepted by the real model. These samples should be non-behavioral samples.

Do the authors still use these sample for global sensitivity analysis?

7. In page 8(step7), all the 500 initial samples are accepted. Since the initial samples are randomly generated without selection, some of them will be non-behavioral samples. Thus, using these samples for global sensitivity analysis will be unreasonable.

8. In page 13, the meta-model selects 4533 samples from 10000 samples. In comparison, only 588 samples from 10000 samples are selected based on the real model. Does this mean the meta-model is not accurate to approximate the real model? Or, do the authors use different criterions to select the behavioral samples for different cases? In addition, the results are based on the behavioral samples obtained by the meta-model. Since there is a significant difference between the obtained behavioral sample by meta-model and the real model, can we really trust the results based on the meta-model? Therefore, a reference result obtained only based on the real model can be provided for comparison.

---

## Referee Comment (RC2) · Anonymous Referee #2 · 26 Jul 2019

General Comments

Overall this paper provides a really nice extension of the active subspaces method to hydrologic modeling. Previous applications of active subspaces to hydrologic models have not used site specific model domains, nor used the active subspaces method in the context of a meta-model. For computationally expensive hydrologic models, the approach presented in this paper may prove to be very valuable to select parameter values but also evaluate sensitvities. The paper is interesting, well-written, and appropriate for this journal.

Specific Comments

1. In order for readers to better understand plots like the ones shown in Figure 4, a description of the active variable needs to be added to the text. Specifically, how the active variable is computed and what the sign (- or +) and magnitude of the active variable means.

2. (Reference Page 8, Line 32) The unsaturated zone (assumed to be between 0 and 1 meter below the land surface in this model) serves as an important link between precipitation falling on the land surface and precipitation reaching the groundwater, and water flowing through this zone can and does contribute to stream discharge. In this model, what is happening to water between 0 and 1 meter below the land surface? How does groundwater recharge occur (Page 12 states that much water is loss to subsurface)? A brief explanation would help especially given that all of the behavioral targets reference stream discharge or recharge.

3. Why does the shape of the Lettenkeuper layer in Figure 1 not match the shapes in Figure 2?

4. Do the magnitude ranges of the output values align with any measured data? For example, and assuming measured data is available, is there some region of the active subspace in Figure 6 that more accurately captures actual watershed behavior?

Technical Corrections

1. Page 8, Line 27 – for clarity, should the word "exit" be replaced with "flow out of the model domain"

2. Figure 1 – main plot should identify the location of Stream A, Stream B, and Gauge C

3. Figure 1 – add acronyms for each geologic layer referenced in Table 1 to Figure 1; for example, Muchelakalk (mo), Lettenkeuper (ku), etc.

4. Figure 1 – Zones 0.5-5.5 are not referenced anywhere in the text of the paper; either add reference or remove legend from Figure 1

5. Page 11, Line 6 – should be flooding not flodding

6. Page 12, Line 5 – for clarity, should "incoming water" be "incoming groundwater", or does incoming water include precipitation and groundwater?

7. Figure 4 – does the behavioral target label "top drain" mean the same thing as "limited flooding" (Section 3.4)? Clarify in the figure description.

8. Figure 4 – the x- and y-axis should be the same for each behavioral target column; this will help the reader compare between differences in the sampling number

No author comments to review or respond to in discussion comments.

―――――――――――――――――

---

## Author Response (AR1)

**Response to Editor and Reviewers**

**Global Sensitivity Analysis and Adaptive Stochastic Sampling of a Subsurface-Flow Model Using Active Subspaces**

hess-2019-163

Daniel Erdal & Olaf A. Cirpka

We would like to thank the editor and the two anonymous reviewers, whose constructive comments have helped improving the manuscript. In the following pages, we will provide detailed answers to each of their comments.

To aid the reading, the original comments by the reviewers are displayed in black, while our replies are both indented and in blue. Italic texts indicate manuscript alterations and references to the revised manuscript are given on the form P8.L9-10, with the meaning page 8 and lines 9-10.

**Editor**

After assessing the Reviewers' comments and the Authors' replies to these, I do think that the study can benefit from a set of moderate revisions. I am indicating "major revisions" to provide the Authors ample time to work on their manuscript.

> We thank the Editor for taking time to handle this manuscript and hope that the alterations done to the manuscript are to his liking!
>
> In summary, these are the three most notable changes implemented in the manuscript upon revision:
>
> 1. Improved description of which samples are used for the post-sampling sensitivity analysis (Reviewer #1, Comments 6-8)
>
> 2. Improved description of the unsaturated zone and the top boundary of the model (Reviewer #2, Comment 2)
>
> 3. Improved motivation with respect to number of subspace dimensions and order of the gradient-approximation polynomials (Reviewer #1, Comments 2 and 4)

**Anonymous Referee #1**

In this paper, the authors used active subspace for global sensitivity analysis and stochastic sampling of a subsurface-flow model. The topic is interesting and suitable for this journal. The adopted method also seems to be efficient and effective. However, some improvements are still required for final publication.

> We thank the reviewer for taking the time to review our manuscript, and for his/her positive view and constructive comments. In the following we address all individual comments one-by-one.

1. There are many useful methods for global sensitivity analysis as stated by the authors in the introduction. Why the authors use the activity score (equation (8))? Can this metric provide more reasonable results for the studied problem than other metrics? Please make an explanation.

> As the reviewer points out, there are different global-sensitivity-analysis methods available, and, despite some existing guidelines, the choice of method is always somewhat subjective to the user. We do not necessarily believe that the activity score provides a more (or less) suitable result than any other method. Our choice of the activity score is primarily based on two reasons.
>
>> 1. We use the active subspace both for sensitivity analysis and as base for a low-order meta-model. This is practical, efficient, and consistent. Other global-sensitivity-analysis methods don't provide a sorted orthogonal basis of parameter vectors that can directly be used to construct a meta-model.
>>
>> 2. To comply with the unstructured sampling performed with the meta-model, we needed a sensitivity analysis which works with random samples, and the activity score is compliant with that (e.g., in contrast to FAST). The relation between the activity score and some existing metrics has been discussed by Constantine and Diaz (2017), but a deeper discussion on that point would be beyond the scope of our paper.
>
> *Manuscript alteration*:
> After introducing the active subspace sampling scheme, the following paragraph has been added (P8.L19-23): *"It should be noted that applying the active-subspace sampling does not necessarily restrict the sensitivity analysis to calculating the activity score (equation 8). As discussed in the introduction, there are many global-sensitivity methods, all with their own strengths and weaknesses, and any method based on a random sample could be applied. In this work, we utilize the fact that we have already computed the active-subspace decomposition so that the post-sampling calculation of the activity score is easy and very cost-effective."*

2. The authors use active subspace to construct a meta-model in order to select the behavioral samples with a lower computational cost. In this work, only 2 active variables are used to construct the meta-model. Why do the authors use 2 active variables? As my understanding, more active variables will make a better meta-model. Do the authors think that 2 active variables are enough to construct an accurate meta-model? Please make an explanation.

The reviewer is correct that in principle more active variables can improve the accuracy of the meta-model. We use two active variables for two main reasons:

1. While the improvement by increasing the number of active variables from one to two was very notable, the improvement by going beyond two active variables was very small.

2. A benefit of using only two active variables is that the full active subspace can be easily visualized, as is seen in the paper. For practical usage, which is a large motivation behind this paper, this is a rather heavy argument.

In summary, we believe that two active dimensions yield enough accuracy in our application, and that the benefit of providing an intuitive visualization outweighs the small benefit of adding more active variables to the sampling scheme.

*Manuscript alteration*:
After the description of the meta-model sampling scheme we have added (P8.L14-19):
*"In this work, we have chosen to construct the meta-model using two active variables. Although more active variables could potentially lead to a higher accuracy of the meta-model, we saw no major improvement when increasing the number of subspace dimensions beyond two. Along the same line of thought, model outcomes in two subspace dimensions can easily be visualized, thus facilitating an intuitive judgment of the goodness of the meta-model. In this light, we retain from going beyond two active subspace dimensions in the current work. Other application may require considering more active dimensions."*

3. The authors use meta-model to select behavioral samples. Since the meta-model only is an approximation of the real model, an error will arise when using meta-model.To construct the meta-model within active subspace, the authors first need to construct an active subspace. This requires the gradients of the model output with respect to the input parameters, which is a computational-demanding task. The authors fit a polynomial to approximate the real model, then obtain the gradient based on the polynomial.This approximation will cause another error. That is to say, there are two kinds of errors caused by approximation (meta-model) in the whole procedure. Since these two kinds of errors are not independent, the final error may be amplified. It will be better to analyze how these two errors will affect the final error of results.

We agree with the reviewer that both approximations introduce errors, and that they are not fully independent of each other. Unfortunately, we do not fully understand what the reviewer suggests us to do. The manuscript already contains a section in which we analyze the error of the gradient approximation (Section 4.3 Gradient Approximation in the Derivation of the Active Subspaces). To separate the active subspace error, however, we would need to have access to correct gradients, which is not possible with the models we work with here. Computing local gradient by direct numerical differentiation at all sampling points would be by far too costly. While we state that we most probably would improve the active subspace with better gradient models (Section 4.3), we cannot distinguish between the error introduced by the gradient approximation and that introduced by the active subspace decomposition.

*Manuscript alteration*:
To increase the clarity in the manuscript, we have added the following paragraph after introducing the active subspace and the gradient approximations (P6.L20-24): *"In summary, the construction of an active subspace contains two strong approximations which both give rise to errors. (1) The active subspace decomposition itself (dimension reduction) and (2) the gradient approximation. As these two errors can be strongly correlated, it is difficult to show the effect of the dimension reduction when the gradient approximation is still uncertain. However, in Section 4.3 we attempt to show the effect on the total error when altering the accuracy of of the gradient approximation."*

4. For the meta-model within active subspace, the authors use third-order polynomial. Why do the authors use third-order? Is third-order enough for the meta-model? Some explanations should be given to convince readers.

The reviewer raises a missing point here. So far, we have taken the standpoint that it is common to use first-order polynomials to estimate the gradients, and hence considered our approach an improvement. However, as the reviewer points out, we have not discussed the upper limit of the polynomial orders. The answer, however, is quite simple, as the number of regression coefficients needed to be estimated grows with the order of the polynomial, and already a third-order polynomial with 32 model parameters has 561 coefficients and, hence, requires already a large set of complete model runs to fit the coefficients. Most flow modeller, who are seen as the main audience of the manuscript, cannot afford such large number of model runs, and that's why we never go beyond a third-order approximation.

*Manuscript alteration*:
Following the description of the gradient approximation, we have added the following new paragraph (P5-6.L28-2): *"In theory, also higher-order polynomials could be used. In practice, however, we are limited by the number of regression coefficients we need to estimate. With the 32 parameters considered in this work, we require a rather large ensemble of full model runs to obtain the 561 b-coefficients, and therefore we retain from considering polynomials above third order."*

5. In page 7(step 2), why do the authors construct an active subspace for each behavioral target? For a scalar model output, there is only one active subspace. Different behavioral targets can be defined based on the same model output. And in this situation, it will be not necessary to construct an active subspace for each behavioral target. I think constructing an active subspace for each model output will be reasonable.

We admit that the formulation was a bit clumsy. Among our behavioral targets, targets 5 and 6 naturally have exactly the same active subspace as they relate to the same observed quantity.

*Manuscript alteration*:
We have changed the text of Step 2 (P7.L14-15) to *"For every unique observation type related to a behavioral target, construct a sufficiently detailed active subspace"*

6. In page 7(step 5), the accepted sample based on the meta-model is used for running the real model and obtain the prediction. Since the meta-model only is an approximation of the real model, the accepted samples based on the meta-model may not be really accepted by the real model. These samples should be non-behavioral samples. Do the authors still use these sample for global sensitivity analysis?

Short answer: no! The only samples used for the sensitivity analysis are those that are deemed behavioral by the real flow model (so called stage-two accepted). The meta-model is only used as a gate-keeper, meant to filter out parameter sets that are highly unlikely to produce behavioral model runs. The final post-sample analysis, however, is done purely on basis of the stage-two accepted real flow models (therefore we ran 10,000 real flow model simulations (stage-one accepted), but only used 4533 out of those for the analysis (stage-two accepted)).

*Manuscript alteration*:
To make this clearer, we have added the following explanations to the manuscript just after the sampling scheme has been described (P8.L9-13)): *"It is important to note, that all post-sampling sensitivity analyses performed in this work are done on the subset of the sampled parameter sets that are deemed behavioral after running the full HydroGeoSphere flow model (stage-two accepted). Hence, we use the meta-model only as a pre-selection tool to avoid sampling those regions of the parameter space that will clearly generate non-behavioral runs. As we aim to sample 10,000 parameter sets (i.e. stage-one accepted), the analyses will be performed on a notably smaller number of parameter sets."*

7. In page 8(step7), all the 500 initial samples are accepted. Since the initial samples are randomly generated without selection, some of them will be non-behavioral samples. Thus, using these samples for global sensitivity analysis will be unreasonable.

This is a misunderstanding. As stated in the answer to comment #6, only samples that are behavioral with respect to the flow model are used for the sensitivity analysis.

*Manuscript alteration*:
Apart from the text to be added mentioned above, we have added the following to step 7 in the description of the sampling scheme (P8.L7-8, addition starts after the comma): *"plus 500 initial samples, which are per-se stage-one accepted"*. We hope this makes it clear that the initial 500 samples are only stage-one accepted before the flow model is run.

8. In page 13, the meta-model selects 4533 samples from 10000 samples. In comparison, only 588 samples from 10000 samples are selected based on the real model. Does this mean the meta-model is not accurate to approximate the real model? Or, do the authors use different criteria to select the behavioral samples for different cases? In addition, the results are based on the behavioral samples obtained by the meta-model. Since there is a significant difference

between the obtained behavioral sample by meta-model and the real model, can we really trust the results based on the meta-model? Therefore, a reference result obtained only based on the real model can be provided for comparison.

There seems to be a misunderstanding here. The 10,000 parameter sets drawn by using the active subspace are assumed to be behavioral by the active subspace meta-model (stage-one accepted). Out of those, 4,533 are actually behavioral after running the real flow model (stage-two accepted), and only those 4,533 parameter sets are used for the analysis. Hence, all results presented in this manuscript are based on the real model and not the meta-model. The comparison with the 588 samples was done to show the effect of using a pure Monte Carlo sampling directly in the full 32-dimensional parameter space (an attempt we did prior to applying the active subspace sampling). We see that this was not clearly enough explained and the manuscript has been be updated to improve this.

*Manuscript alteration*:
Apart from the answer to comments #6 and #7, as well as adding throughout the result-section the terminology stage-one and stage-two acceptance (please see the track-change document), we have altered the description of sampling the 588 comparison samples to (P14.L23-26): *"In a comparable setup, we sampled 10,000 parameter-sets using a pure Monte Carlo sampling scheme (i.e. without any kind of meta-model or pre-selection), and out of those, only 588 were acceptable with the strict criteria used here (i.e. stage-two accepted, although in the case of a pure Monte-Carlo sampling, no stage-one acceptance is tested). Hence, the improvement when using the active subspace sampler is clearly notable."*

In summary, we hope that the changes provided as responses to comments #6-8 makes it clear how both the sampling and the subsequent sensitivity analysis is done in our work.

**Anonymous Referee #2**

**General Comments**

Overall this paper provides a really nice extension of the active subspaces method to hydrologic modeling. Previous applications of active subspaces to hydrologic models have not used site specific model domains, nor used the active subspaces method in the context of a meta-model. For computationally expensive hydrologic models, the approach presented in this paper may prove to be very valuable to select parameter values but also evaluate sensitvities. The paper is interesting, well-written, and appropriate for this journal.

> We would like to thank the reviewer for his/her positive summary of our manuscript.

**Specific Comments**

1. In order for readers to better understand plots like the ones shown in Figure 4, a description of the active variable needs to be added to the text. Specifically, how the active variable is computed and what the sign (- or +) and magnitude of the active variable means.

> We admit that the description of the active variable was rather short. An active variable is a linear combination of several flow-model parameters. Because the active variables are linear combinations, the absolute values and signs of the active variables don't have an immediate intuitive meaning. In order to relate values of active dimensions to physical parameters, one needs to look up the exact eigenvectors used to construct the active variable (available in the supplementary material), or, as we have done in this manuscript, look at the activity score.
>
> *Manuscript alteration*:
> We have clarified this in the revised manuscript by the following new paragraph (P14.L29-33, in the results section related to Figure 4): *"Each of the two active variables (equation 12) is a linear combination of flow-model parameters, weighted by their respective influence on the specific subspace dimension. Due to its construction, the active variable itself is hard to interpret. However, the activity score (equation 8), used in this work to judge the importance of the physical parameters, effectively shows the components of the active variable, and is therefore the preferred way to interpret the parameter-related results."*

2. (Reference Page 8, Line 32) The unsaturated zone (assumed to be between 0 and 1 meter below the land surface in this model) serves as an important link between precipitation falling on the land surface and precipitation reaching the groundwater, and water flowing through this zone can and does contribute to stream discharge. In this model, what is happening to water between 0 and 1 meter below the land surface? How does groundwater recharge occur (Page 12 states that much water is loss to subsurface)? A brief explanation would help especially given that all of the behavioral targets reference stream discharge or recharge.

We see that this part was not so well explained in the original submission, and we thank the reviewer for pointing this out. First of all, it is only the first subsurface meter that is missing in the model, not the full unsaturated zone. On the contrary, the largest area of the domain has a notable unsaturated zone. The missing top meter is where intricate plant-soil-atmosphere feedbacks occur. We do not wish to model this explicitly for the sake of simplicity. Instead, we use simulation results from independent one-dimensional crop models that consider the mentioned feedbacks to obtain the mean and standard deviation of the volume flux at one-meter depth, depending on the land-use type. These fluxes are then assigned as Neumann boundaries, which corresponds to the local groundwater recharge (thus neglecting any potential interflow). Water can leave the domain through the land surface, when the pressure head in the upper-most nodes of the model domain is higher than a reference pressure head of 1.2 meter (i.e. 1 meter to "reach the surface" and 20 cm of actual ponding).

*Manuscript alteration*:
To make this clearer, the description of the upper boundary conditions has been updated as follows (P10.L12-20): "To avoid long run times and complications of complex top soils *(including plant-atmosphere interactions)*, which are unimportant once steady state is reached, the top of the HydroGeoSphere model is 1 meter below the land surface. Flow across the top boundary is only incoming and modelled as a Neumann-boundary, corresponding to the steady-state groundwater recharge. The recharge varies with land-use, split into three categories: cropland, grassland, and forest, in which urban areas are treated as grassland. *It should be noted that starting the model at 1 meter below surface still allows for a notable unsaturated zone to develop in the model domain; only the upper-most meter is missing. Also, the outgoing drain-fluxes described above are applied to this one-meter-below-surface top boundary, hence requiring that the exit pressure head is one meter larger than the ponding pressure-head described above. Technically, this does not apply to to stream nodes, which are considered to be the real top of the porous medium (that is, there is no unsaturated zone on top of a stream).*"

3. Why does the shape of the Lettenkeuper layer in Figure 1 not match the shapes in Figure 2?

In our work, the shape of the Lettenkeuper layer (like that of all other layers) is controlled by a combination of stochastic input parameters. Figures 1 and 2 show examples of random realizations, and it happens that the 6 realizations shown in Figure 2 do not include the realization shown in Figure 1.

*Manuscript alteration*:
At the point where Figure 2 is introduced, we have added the following sentence (P11.L15-16): *"Please note that Figure 2 shows different realizations than Figure 1."*

4. Do the magnitude ranges of the output values align with any measured data? For example, and assuming measured data is available, is there some region of the active subspace in Figure 6 that more accurately captures actual watershed behavior?

The reviewer raises an important point. Indeed, the output values used as targets for the sampling scheme are already setup to reflect realistic catchment behavior. Although not based on hard data, the targets specifies ranges that most hydrologists familiar with the catchment would find very reasonable. Hence, all results shown in active subspaces, e.g. in Figure 6, are in line with measured data. Of course, if better data were available, it would be possible to identify sub-regions in the active subspace that matches those data. However, it should be noted that although it is easy to find the region in active subspace, the back-transform to parameter space is non-unique, and we hence need the sampling to infer real parameter ranges.

*Manuscript alteration*:
To clarify this, we have added the following paragraph just after the targets are introduced (P13.L4-9)): *"As all targets are based on knowledge about the real catchment on which the current model is based, all model outputs produced by a stage-two accepted model will be in line with what we would expect to be realistic in the catchment. However, it is important to note that even though it would be possible to point out a location in the active subspace which corresponds to a real observation, the active variables themselves are non-unique with respect to the flow parameters, so that a simple back-transformation from active subspace to flow-model parameter space is not possible."*

**Technical Corrections**

1. Page 8, Line 27 – for clarity, should the word "exit" be replaced with "flow out of the model domain"

Altered as suggested (P10.L6)

2. Figure 1 – main plot should identify the location of Stream A, Stream B, and GaugeC

This information disappeared in the revision of the figure, we thank the reviewer for catching this error!

3. Figure 1 – add acronyms for each geologic layer referenced in Table 1 to Figure 1;for example, Muchelkalk (mo), Lettenkeuper (ku), etc.

This is a good suggestion and has been included

4. Figure 1 – Zones 0.5-5.5 are not referenced anywhere in the text of the paper; either add reference or remove legend from Figure 1

Altered as suggested

5. Page 11, Line 6 – should be flooding not flodding

Typo corrected

6. Page 12, Line 5 – for clarity, should "incoming water" be "incoming groundwater", or does incoming water include precipitation and groundwater?

Altered as suggested (P12.L12)

7. Figure 4 – does the behavioral target label "top drain" mean the same thing as "limited flooding" (Section 3.4)? Clarify in the figure description.

Altered as suggested for both Figure 4 and Figure 5 (see page 16 in revised manuscript)

8. Figure 4 – the x- and y-axis should be the same for each behavioral target column;this will help the reader compare between differences in the sampling number.

We actually find this suggestion a bit misleading, as the composition, and hence the values, of the active variables can change with increasing data volume (see answer to comment #1 and the differences seen in Figure 4 where the middle columns two upper plots are differently oriented). Using the same axis would give the misleading impression that the active variables are always the same and only stabilize with more data. Hence, we have made no changes following this comment.

[revised manuscript text omitted]

 It is important to note, that all post-sampling sensitivity analyses performed in this work are done on the subset of the sampled parameter sets that are deemed behavioral after running the full HydroGeoSphere flow model (stage-two accepted). Hence, we use the meta-model only as a pre-selection tool to avoid sampling those regions of the parameter space that will clearly generate non-behavioral runs. As we aim to sample 10, 000 parameter sets (which are stage-one accepted), the analyses will be performed on a notably smaller number of parameter sets.

In this work, we have chosen to construct the meta-model using two active variables. Although more active variables could potentially lead to a higher accuracy of the meta-model, we saw no major improvement when increasing the number of subspace dimensions beyond two. Along the same line of thought, model outcomes in two subspace dimensions can easily be visualized, thus facilitating an intuitive judgment of the goodness of the meta-model. In this light, we retain from going beyond two active-subspace dimensions in the current work. Other application may require considering more active dimensions.

It should be noted that applying the active-subspace sampling does not necessarily restrict the sensitivity analysis to calculating the activity score (equation 8). As discussed in the introduction, there are many global-sensitivity methods, all with their own strengths and weaknesses, and any method based on a random sample could be applied. In this work, we utilize the fact that we have already computed the active-subspace decomposition so that the post-sampling calculation of the activity score is easy and very cost-effective.

**3 Application to a Virtual Test Case**

**3.1 Description of the Domain**

The test bed used in this paper is a steady-state flow model setup and run in HydroGeoSphere. It draws its main features from the catchment of the stream Käsbach in the Ammer-valley in southwestern Germany (Selle et al., 2013) with some simplified features. That is, the simulated domain is not meant as exact representation of the Käsbach catchment but contains enough details to be considered a realistic test for the proposed global sensitivity analysis method.

[Figure]

**Figure 1.** Illustration of the modelled catchment, including important features and surrounded by an explicit view on the different geological features.

As illustrated in Figure 1, the subsurface model consist of 5 geological layers, representing the major lithostratigraphic units in the region. From the bottom to the top, these are: (1) the middle-Triassic Upper Muschelkalk formation, made of fractured-karstified limestone, (2) the lower upper-Triassic Lettenkeuper (Erfurt formation), made of clay-rich mudstones and carbonate-rock layers, (3) the unweathered middle upper-Triassic Gipskeuper (Grabfeld formation), made of mudstones and gypsum-bearing layers, (4) a weathering zone of the latter formation, and (5) Quaternary valley fills of unconsolidated sediments. A fault passes through the domain in the North-South direction, leading to offsets in the geological units. The geological base model resembles the regional model of D'Affonseca et al. (2018). Each layer is modeled as homogeneous unit.

The model domain measures about 4 km × 6 km at the widest places. It is discretized by 1,001,760 prism elements using 523,083 nodes and features a single main stream with four possible tributaries. The model is setup and run in transient mode with constant forcings until steady state is reached. Only the final time step (here after simulating $10^{10}$ seconds) is considered in the analysis.

The boundary conditions, setup to allow water to leave the domain both through the surface and the subsurface, are as follows: The bottom of the model features a Dirichlet-boundary with values read in from a larger-scale model of the region (D'Affonseca et al., 2018), but limited such that no hydraulic head at the bottom face can be higher than 5 meters below the model top. Streams in the model are modelled as drains, meaning that water can  flow out of the domain when the hydraulic head at the assigned stream nodes exceeds a value 1 cm above the surface elevation. This implies that all streams are either inactive or gaining, whereas losing conditions are excluded. A similar drain boundary, but with much higher exit head (fixed at 0.2 meters above land surface for all simulations) is also considered on all non-stream nodes in the upper most layer, to allow water to leave the domain in case of flooding. The last outflow boundary in the model is a Cauchy-boundary at the southern vertical wall of the model.

To avoid long run times and complications of complex top soils (including plant-atmosphere interactions), which are unimportant once steady state is reached, the top of the HydroGeoSphere model is 1 meter below the land surface. Flow across the top boundary is only incoming and modelled as a Neumann-boundary, corresponding to the steady-state groundwater recharge. The recharge varies with land-use, split into three categories: cropland, grassland, and forest, in which urban areas are treated as grassland. It should be noted that starting the model at 1 meter below surface still allows for a notable unsaturated zone to develop in the model domain; only the upper-most meter is missing. Also, the outgoing drain-fluxes described above are applied to this one-meter-below-surface top boundary, hence requiring that the exit pressure head is one meter larger than the ponding pressure-head described above. Technically, this does not apply to to stream nodes, which are considered to be the real top of the porous medium (that is, there is no unsaturated zone on top of a stream).

[revised manuscript text omitted]
. As all targets are based on knowledge about the real catchment on which the current model is based, all model outputs produced by a stage-two accepted model will be in line with what we would expect to be realistic in the catchment. However, it is important to note that even though it would be possible to point out a location in the active subspace which corresponds to a real observation,

the active variables themselves are non-unique with respect to the flow parameters, so that a simple back-transformation from active subspace to flow-model parameter space is not possible.

**4    Results**

To allow the reader to better see the 3-D  structure in the results presented here, the main results can be viewed in a
5    plug-and-play app designed for Matlab, denoted "Active Subspace Pilot", which is available as supplementary information to this publication.

**4.1    Adaptive Sampling**

The effect of using the active subspaces as a sampling strategy for the flow simulations can be seen in Figure 3, showing the marginal distributions of the nine parameters that were most influenced by the sampling strategy. The blue bars are histograms
10    of all parameter sets selected for full model runs, whereas the red bars are histograms of the behavioral parameter sets.

The blue bars of Figure 3 clearly show that already the pre-selection using the meta-model avoids certain regions of the parameter space. In particular, the two parameters related to the weathering zone of the Gipskeuper (conductivity and thickness of km1-w in Figure 3) show a preferential sampling for a thick and highly conductive layer. Similar preferences are seen in the Lettenkeuper and the Gipskeuper (ku and km1). This is contrasted by the deeper subsurface, where the Muschelkalk (mo)
15    shows a preference towards low conductivity and the offset of the fault is preferably sampled at smaller values (which decreases the size and connectivity of the Muschelkalk layer). By selecting high conductivity values in the Quaternary and weathered Gipskeuper layers, chances of floodings are reduced. Further, the high conductive near-surface and middle-deep layers serve to transport water towards the streams. The smaller and less conductive deep subsurface in combination with higher bottom pressures, on the other hand, serves to inhibit exiting water through the bottom. Hence, the posterior sample shows exactly the
20    behaviors required by the targets. This suggests that the sampling strategy has been successful.

The red bars in Figure 3 show the marginal posterior distribution of parameters used for the sensitivity analysis (stage-two accepted parameter sets). This corresponds to a selection of simulations that are strictly better than the mean of the targets and their outer point (see above). Hence, this selection is deterministic with a hard limit. It is obvious that the pre-selection (stage-one acceptance) and final selection (stage-two acceptance) of parameters are similar, but the stricter sample used in the
25    sensitivity analysis has fewer members. The blue bars in Figure 3 comprises 10,000 samples (stage-one acceptance), while the red bars comprises a subset of 4,533 samples (stage-two acceptance). Part of the reason for this rather larger difference, is that the active subspace sampler is only an approximation. More so, we have deliberately relaxed the criterion for accepting a parameter set by including a range around the target, and, for the full run to include a 10%  pre-acceptance independent of the meta-model prediction. In a comparable setup, we sampled 10,000 parameter-sets  using a pure Monte-Carlo
30    sampling scheme (i.e. without any kind of meta-model or pre-selection), and out of those, only 588 were acceptable with the strict criteria used here (i.e. stage-two accepted, although in the case of a pure Monte-Carlo sampling, no stage-one acceptance is tested). Hence, the improvement when using the active subspace sampler is clearly notable.

[Figure]

**Figure 3.** Marginal posterior distributions of the parameters influenced by the adaptive sampling. Blue bars show the sampled posterior and red bars the constrained posterior sample used in the sensitivity analysis.

Figure 4 shows the performance and development of the active subspaces for three representative targets. Here, the  $x$- and $y$-positions of the markers are the values of the two active variables, respectively, while the color indicates the magnitude of the corresponding observation. Each of the two active variables (equation 12) is a linear combination of the flow-model parameters, weighted by their respective influence on the specific subspace dimension. Due to its construction, the active variable itself is hard to interpret. However, the activity score (equation 8) used in this work to judge the importance of the physical parameters, effectively shows the components of the active variable, and is therefore the preferred way to interpret the parameter-related results. 
[revised manuscript text omitted]